# Left Ventricular Adverse Remodeling in Ischemic Heart Disease: Emerging Cardiac Magnetic Resonance Imaging Biomarkers

**DOI:** 10.3390/jcm12010334

**Published:** 2023-01-01

**Authors:** Camilla Calvieri, Alessandra Riva, Francesco Sturla, Lorenzo Dominici, Luca Conia, Carlo Gaudio, Fabio Miraldi, Francesco Secchi, Nicola Galea

**Affiliations:** 1Department of Clinical, Internal, Anesthesiologic and Cardiovascular Sciences, Sapienza University of Rome, 00100 Rome, Italy; 2Department of Electronics, Information and Bioengineering, Politecnico di Milano, 20129 Milan, Italy; 33D and Computer Simulation Laboratory, IRCCS Policlinico San Donato, 20097 Milan, Italy; 4Department of Radiological, Oncological and Pathological Sciences, Sapienza University of Rome, 00100 Rome, Italy; 5Unit of Radiology, IRCCS Policlinico San Donato, 20097 Milan, Italy; 6Department of Biomedical Sciences for Health, Università Degli Studi di Milano, 20129 Milan, Italy

**Keywords:** adverse remodeling, left ventricular remodeling, myocardial infarction, cardiac magnetic resonance imaging, 4D flow, feature-tracking myocardial strain, surgical ventricular restoration

## Abstract

Post-ischemic left ventricular (LV) remodeling is a biologically complex process involving myocardial structure, LV shape, and function, beginning early after myocardial infarction (MI) and lasting until 1 year. Adverse remodeling is a post-MI maladaptive process that has been associated with long-term poor clinical outcomes. Cardiac Magnetic Resonance (CMR) is the best tool to define adverse remodeling because of its ability to accurately measure LV end-diastolic and end-systolic volumes and their variation over time and to characterize the underlying myocardial changes. Therefore, CMR is the gold standard method to assess in vivo myocardial infarction extension and to detect the presence of microvascular obstruction and intramyocardial hemorrhage, both associated with adverse remodeling. In recent times, new CMR quantitative biomarkers emerged as predictive of post-ischemic adverse remodeling, such as T1 mapping, myocardial strain, and 4D flow. Additionally, CMR T1 mapping imaging may depict infarcted tissue and assess diffuse myocardial fibrosis by using surrogate markers such as extracellular volume fraction, which may predict functional recovery or risk stratification of remodeling. Finally, there is emerging evidence supporting the utility of intracavitary blood flow kinetic energy and hemodynamic features assessed by the 4D flow CMR technique as early predictors of remodeling.

## 1. Introduction

Left ventricular (LV) remodeling after myocardial infarction (MI) is the process clinically manifesting as a change in LV architecture, size, and function, regulated by hemodynamic load, neuro-hormonal activation, and genetic factors, which begins within the first hours after MI and lasts up to 1 year [1,2]. Histologically, it is driven by a combination of myocyte hypertrophy and apoptosis, myofibroblast proliferation, and interstitial fibrosis and plays a pivotal role in the development of heart failure (HF) [3,4,5].

LV remodeling can be distinguished into two types: the first, physiological and adaptive during development, and the second, maladaptive, as a result of several pathophysiological mechanisms (i.e., ischemic heart disease, cardiomyopathy, hypertension, valve dysfunctions) [5,6]. The pathophysiology of post-ischemic LV remodeling is complex, with multiple ultrastructural, metabolic, and neurally mediated processes occurring in infarcted and remote myocardium [5,7,8]. Cardiomyocyte necrosis and the resulting increased overload of the remaining fibers trigger a cascade of biochemical intracellular signaling processes that initiate and modulate reparative changes such as dilation, hypertrophy, and fibrosis [1]. The initial phase (within 72 h) involves the dilation and thinning of the infarcted wall, which can cause premature rupture or aneurysmal deformation and facilitate intracavitary thrombus formation [9,10,11]. The acute loss of myocardium after infarction causes a sudden increase in wall shear stress and chamber dilation, which involves the tissue bordering the necrotic area and the remote myocardium [1]. Further, late LV remodeling (beyond 72 h) affects the global cavity and is associated with time-dependent changes of cardiac geometry, from an elliptic to a spheric shape, and chamber enlargement, ultimately resulting in HF [3,12].

Despite early reperfusion strategies, nearly 30% of ST-elevation myocardial infarction (STEMI) patients present a higher risk of developing LV adverse remodeling at mid-term FU, which is associated with a higher incidence of long-term poor outcomes such as cardiovascular death, MI, stroke, hospitalization for HF, and resuscitated cardiac arrest [13]. However, there is no univocal definition of LV adverse remodeling due to the heterogeneous thresholds used by the different imaging techniques, initially obtained by echocardiography and then re-defined by cardiac magnetic resonance (CMR) imaging.

Adverse LV remodeling in post-ischemic patients has been defined firstly by echocardiography as Δ20% increase in LVEDV 6 months after STEMI, on the basis of the upper limit of the 95% confidence interval of intraobserver variability for change (%Δ) in LVEDV [14].

Echocardiography remains an essential tool to characterize adverse remodeling by using two-dimensional (2D) and three-dimensional (3D) analysis and to predict functional recovery by speckle tracking (2D-STE), whereas contrast echocardiography improves the detection of thrombi in the left ventricle. However, cardiac multimodality imaging allows for the definition of different structural and functional aspects of remodeling, especially for the assessment of LV volumes and geometry [15]. In this regard, multidetector computed tomography (MDCT) has been shown to be capable of providing highly reproducible cardiac volume measurements and function assessments compared to CMR [16]. Meanwhile, nuclear imaging with radiolabeled biomolecules has been used to non-invasively quantify activation of myocardial fibroblasts [17] and it can be employed for assessment and detection of LV remodeling even though it is not routinely included in clinical practice [18]. Among these imaging techniques, CMR is the most indicated tool to define LV remodeling, due to the high reproducibility of cardiac volumes and function assessment. Moreover, CMR enables to quantify infarct size (IS), detect microvascular obstruction (MVO), and detect intramyocardial hemorrhage (IMH), representing key determinants of adverse remodeling and predictors of poor cardiovascular outcomes [19,20]. MDCT is also able to detect the infarcted myocardium and MVO using late iodine enhancement (LIE) imaging, providing information for the early prediction of adverse LV remodeling [21]. Although MDCT offers higher spatial resolution and a shorter scan time, its widespread adoption is limited by the need for adequate heart rate control and a heavy radiation burden. The superior capability of CMR compared to MDCT for soft tissue characterization also favors the former approach [22]. Thus, our aim was to describe the different roles of established and new CMR imaging techniques in characterizing and defining the adverse remodeling in the different phases of progression.

## 2. LV Remodeling Volume-Based Definition

The importance of defining post-MI LV remodeling comes from several CMR studies that demonstrated its association with worse cardiovascular outcomes [23] and, conversely, the effect of reverse remodeling on improving the long-term prognosis [2,24]. CMR and 2D transthoracic (TTE) echocardiography are the principal techniques for volume assessment. If compared with CMR, 2D TTE echocardiography reproduces smaller volumes and larger masses [25]. Further, LV volumes and function reproducibility can be influenced by the imaging technique used, but agreements are lower than 5% with CMR [26]. 2D echocardiography seems to underestimate LVEF in non-anterior STEMI if compared with CMR, whereas after anterior STEMI no significant difference in LVEF measurements by 2D echo and CMR was found [27]. In STEMI patients, it has been demonstrated that CMR LVEF is predictive of MACE only in the group with echocardiography-LVEF < 50% [28]. Otherwise, 3D echocardiography is more precise than 2D echo to obtain LV volumes and function in STEMI patients, being independent from geometrical assumptions and unaffected by foreshortening, especially in cases of distorced LV shape. Additionally, 3D echocardiography is more accurate for volume assessment, measuring LVEF [29] and detecting adverse LV remodeling compared to 2D echo [30]. CMR is the gold standard not only to evaluate myocardial tissue characterization but also for measuring LV volumes, LV ejection fraction (EF), and myocardial mass. The main criteria for the quantification of LV remodeling after MI are the relative percentage changes of LV end-diastolic (%ΔEDV) and end-systolic (%ΔESV) volumes between baseline and follow-up (FU) CMR exams [31]. Thus, different LV remodeling cut-off values, as well as %ΔEF and %ΔMass changes, have mainly been considered during a follow-up period of 4–6 months. The most frequently used thresholds are %ΔEDV > +20% or %ΔESV > +15%, derived by echocardiography, as previously mentioned [15,31,32]. However, there is still no consensus on volume thresholds to define adverse LV remodeling by CMR. Furthermore, in several studies, different FU periods, ranging from 4 to 8 months between the first CMR and the FU CMR, have been considered to define the changes in LV volumes. Additionally, Reindl et al. defined LV adverse remodeling as a %ΔEDV ≥ 10% at 4 months’ FU as the best predictor of 24 months’ cardiovascular events (MACEs: stroke, non-fatal myocardial infarction reinfarction, new congestive HF, and all-cause mortality) [24,33]. Rodriguez Palomares et al. defined adverse remodeling, combining a >15% increase in EDV with a 3% EF reduction at 6 months, as the best predictor of 6-year cardiovascular events [23]. Recently, Bulluck et al. suggested new LV remodeling categories, on the basis of a 5-year composite end point of all-cause death and hospitalization for HF, by using a cut-off of 12% for both %ΔEDV and %ΔESV between baseline and 6 months FU CMR exams, identifying 4 principal patterns of LV remodeling: reverse LV remodeling (≥12% decrease in %ΔESV), no remodeling (changes in %ΔEDV and %ΔESV < 12%), adverse remodeling with compensation (≥12% increase in %ΔEDV only), adverse remodeling (≥12% increase in both %ΔESV and %ΔEDV) [34]. They demonstrated that the latest group with both a %ΔEDV and %ΔESV increase of 12% at 6 months was associated with a higher risk of the composite end point of all causes of mortality and HF during a median of 5 years with a Hazard ratio (H.R.) of 3.0 (95% CI 1.2–7.2), as compared to the other 3 groups. Meanwhile, reverse remodeling was often defined as a decrease of >10% in ESVi at 6-months post-MI CMR imaging [2]. Regarding %ΔMass cutoffs, Bulluck et al. identified a minimal change in %ΔMass of 12% between acute and follow-up scans as the cutoff not attributable to inter-observer measurement error. Recently, Xu et al. found that a %ΔMass ≥ 15% at 1 year was the best predictor, among other parameters of cardiac remodeling, of a primary outcome such as death or cardiovascular hospitalization, independently from LVEF and LVEDV [35]. Reindl et al. reported an optimal cutoff %ΔMass of ≥5% to predict 2-year MACE, incrementing the prognostic value of %ΔEDV ≥ 10% [24]. However, changes in LV mass could be influenced by the presence of acute myocardial edema during the follow-up period, so many studies still only consider changes in EDV or ESV as the cut-offs for defining LV adverse remodeling.

## 3. Myocardial Strain Analysis

Myocardial strain consists of the deformation of the ventricular wall due to longitudinal and circumferential shortening, radial thickening, and torsion [36]. Myocardial deformation imaging using 2D STE allows quantification of regional myocardial function and of LV longitudinal motion and is superior to LVEF for predicting adverse remodeling and poor outcomes [37]. The 2D STE is derived from the backscatter of ultrasound within myocardial tissue. In addition, strain analysis, obtained by advanced post-processing of cine-CMR images by dedicated software, may improve the evaluation of global and regional systolic and diastolic function [36,38]. A major limitation of 2D STE is image quality, especially in patients with poor imaging windows, ultrasound dropouts, and increased field noise [39]. Thus, CMR-derived strain analysis can be useful in those patients with difficult echogenic windows and offers superior image quality, as demonstrated by the fewer non-analysable segments and larger fields of view [39]. The CMR feature-tracking (FT) analysis allows to quantify myocardial wall deformation, independently from ventricular size or shape, without the need for additional sequences, which are necessary when the strain is measured using tissue-tagged cine sequences [40].

Among myocardial strain parameters, LV global longitudinal strain (GLS) has been frequently identified as the best predictor of adverse remodeling with different thresholds, while its reduction has been associated with long-term poor cardiovascular outcomes [33,41]. In STEMI patients, lower CMR-FT GLS values have been reported if compared with STE GLS, despite a good agreement in GLS quantification by both CMR-FT and 2D STE, whereas only moderate agreement in segmental analysis is due to worse basal segment tracking [42]. In a multilayer GLS assessment study, 2D STE tends to overestimate endocardial strain and underestimate epicardial strain if compared to CMR-FT [43]. In a large multicentric study of 1235 patients with MI (both STEMI and non-ST elevation MI), Eitel et al. demonstrated that those showing MACEs at 1 year after infarction had lower GLS values compared to those without MACEs [44]. Additionally, GLS showed an additive prognostic value over EF in predicting all-cause mortality with a significant increase in the c-statistic (for model EF + GLS, AUC: 0.73; 95% CI: 0.70 to 0.76; *p* = 0.04). Cha et al. [41] confirmed that GLS is a predictor of LV remodeling (O.R. 1.282, 95% CI 1.060–1.550, *p* = 0.011) with an optimal cut-off of −12,84, independently from IS, IMH, and MVO. In the paper of Garg et al., baseline GLS was closely associated with MVO and IMH, with an optimal cut-off of −13.7% (sensitivity 76%, specificity 78%), and it was the strongest predictor of LV adverse remodeling with an AUC of 0.79 (95% CI 0.60–0.98, *p* = 0.03) [45]. However, GLS may be influenced by myocardial infarct and non-infarct-related segments because hyperkinetic non-infarcted myocardial segments could restore the global longitudinal contractility, leading to normal GLS values in some patients [46].

Furthermore, the reduction of global circumferential strain (GCS) has been demonstrated to predict adverse LV remodeling [47]. The multilayer 2D STE GCS may add diagnostic value in assessing infarct transmurality and predicting functional recovery in post-ischemic patients [48]. In particular, the 2D STE reduction in subendocardial strain has been demonstrated to anticipate ischemic changes [49]. In CMR FT, Buss et al. described a strict association between GCS and both IS and transmurality of the scar, demonstrating that GCS with a cut-off of 19.3% is the best predictor (i.e., sensitivity and specificity equal to 56% and 85%, respectively) of preserved EF (>50%) at 6 months in a cohort of 74 reperfused STEMI patients [50]. Additionally, Holmes et al. [51] found that GCS was more predictive of LGE in determining myocardial segments undergoing late adverse remodeling and LV dysfunction at 3 months [51]. Interestingly, they demonstrated that GCS was superior in defining the degree of transmurality of the infarcted area and that the extent of impaired GCS may be associated with implantable cardioverter defibrillator (ICD) therapy [51]. Paiman et al. confirmed, in a cohort of 121 patients with ischemic cardiomyopathy, the role of GCS in predicting appropriate ICD therapy, independently from EF, scar extension, and coronary revascularization [52]. Recently, we demonstrated that GCS and GLS have high specificity to detect 4 months of adverse CMR remodeling when compared to other CMR indices of MI characterization such as MVO, IMH, and IS [47]. Thus, the measurement of myocardial strain could provide important additional information on the development of LV dysfunction and adverse remodeling.

## 4. Tissue Characterization

### 4.1. LGE, MVO, and IMH

As a cornerstone of CMR imaging of MI, late gadolinium enhanced imaging (LGE) is the established reference standard technique for the in vivo assessment of myocardial viability, providing an excellent depiction of the infarcted necrotic myocardium during the subacute phase and the fibrotic scar following the healing process [53]. Several studies showed that the IS, assessed as the relative LGE volume on total LV ventricular mass, is a major determinant of LV adverse remodeling [54,55,56,57]. IS is linearly and strongly correlated with LV-ndexed volumes in FU examinations at different time points [58]. In a study by Pezel et al. [57], after multivariate analysis, IS was the major predictor of adverse LV remodeling [54]. Myocardial perfusion imaging assessed by gated single-photon emission computed tomography (SPECT) also found similar results: infarct size and degree of transmurality, expressed as infarct severity ratio, were able to predict remodeling with 86% sensitivity, 72% specificity, and 75% accuracy when considered together [59]. The progressive replacement of the necrotic myocardium by scar tissue is characterized by shrinkage of the damaged tissue, a phenomenon devoted to minimizing the amount of dysfunctional myocardium and counterbalancing the increased tension in the infarcted wall [60].

In CMR images, LGE volume is reduced by about 40% between the early phase and 4 months FU after MI, whereas there is no further significant reduction at the 1-year FU, as demonstrated by studies in patients with acute STEMI undergoing primary PCI [61]. The severity of myocardial injury may also be assessed by the transmurality of myocardial scars, which is estimated by the percentage extent of the infarcted myocardium relative to the entire wall thickness on LGE images. The involvement of all myocardial layers by necrosis leads to more severe contractile impairment and worse functional recovery at FU. As demonstrated in a previous study [62], subepicardial fiber shortening improved during FU and was responsible for the partial and progressive recovery of contractile function, compensating for the mid-subendocardial layer damage. Consequently, the infarct thickness had a significant effect on the degree of local remodeling at one year, with greater wall dilation observed when the infarct involved more than 50% of LV wall thickness [63]. A further study also demonstrated that the number (per patient) of LV segments with transmural necrosis had additional predictive value for early LV remodeling and worse LV ejection fraction, independently of severe MVO and IS [55]. Infarction’s location also plays a role in LV remodeling. Anterior infarcts are more likely to lead to adverse remodeling than inferior or lateral, even though this is likely attributable to the larger IS when the left anterior descending artery is involved without any independent contribution of infarct location per se [54,58].

An important role of LGE in patients with post-ischemic heart failure is also related to the stratification of patients with an indication for conventional cardiac resynchronization therapy (CRT). In the literature, it is reported that a transmural LGE in the posterolateral wall is related to nonresponse to CRT. As a result, this evidence shows that CRT does not reduce LV dyssynchrony in patients with transmural scar tissue in the posterolateral LV segments, resulting in clinical and echocardiographic nonresponse to CRT even if extensive LV dyssynchrony exists [64]. A transmural scar in the target region (for LV pacing) prohibits response to CRT [65].

MVO consists of a dark zone within the LGE areas as a result of the “no-reflow” phenomenon induced by coronary reperfusion therapy following prolonged myocardial ischemia. Essentially, it is related to the damage and dysfunction of the myocardial microcirculation through the combination of vasoconstriction and endothelial swelling associated with myocardial cellular edema, hindering blood flow to penetrate beyond the myocardial capillary bed despite revascularization [66,67].

IMH is closely related to MVO, being due to the interstitial extravasation of red blood cells in the myocardium as a result of the disruption of the microcirculation after ischemia-reperfusion injury [68,69]. Further, IMH can be assessed as a hypointense core inside the area of post-infarction myocardial edema on T2-weighted or T2 * images, exploiting the paramagnetic effects of hemoglobin degradation products and iron deposition [69]. The amount of collateral flow, ischemic preconditioning, and extent of necrosis also correlate with the presence and severity of IMH [70]. The presence of MVO and IMH, as markers of severe myocardial reperfusion injury, was proven to be an independent predictor of post-infarction LV adverse remodeling and poor clinical outcome following MI [71,72,73,74,75,76]. In a study conducted by Baks et al. [77], myocardial segments with MVO showed wall thinning at five months after STEMI and no significant recovery of function compared with remote segments. The presence of late MVO, rather than its size, is considered to be the strongest predictor of increased ESV, EDV, and poor EF at FU, representing a surrogate marker of LV remodeling [78]. A similar conclusion was drawn by other studies [61,79] that proved MVO as an independent predictor of adverse LV remodeling regardless of IS. Additionally, MVO affects the repair processes on infarcted myocardium, which result in three distinct patterns of LV remodeling at FU: normal healing, dilation with functional adaptation, and dilation without adaptation [79]. In a study from Orn et al. [79], in a cohort of 42 STEMI patients evaluated at different time points of 2 days, 1 week, and 1 year after PCI, MVO at 1 week was an independent predictor of IS, poor EF, increased LV volumes, and adverse LV remodeling at 1 year FU. In addition, the presence of IMH is associated with a larger IS, a larger MVO, a lower LVEF, and adverse LV remodeling [80]. The CMR gradient-echo T2 * sequence has been demonstrated to improve the sensitivity for IMH in STEMI patients [81].

The presence of heterogeneous LIE and high relative myocardial density on MDCT scans performed 5–10 min after contrast administration have been considered an early predictor of MVO and subsequent LV remodeling [21].

The optimal timing for performing CMR after STEMI remains unclear. A large multicentre CMR cohort study (CoReCMR-in-STEMI) demonstrated that either an early CMR strategy (within 6 days from reperfusion), a deferred CMR strategy (within 9 months from the acute event), or a paired CMR strategy (by the combination of CMR parameters with LV remodeling parameters) were similar in predicting all causes of death and HF hospitalization during long-term follow-up [20]. An early CMR strategy is still widely debated due to histopathological changes during the acute phase post-STEMI, mainly represented by a significant overestimation of IS and LV adverse remodeling when CMR is performed during the first week after STEMI. Additionally, if LGE decreases within the first week and then remains stable for 4 months after STEMI shows a 50% resolution by 6 months, myocardial oedema is constant over the first week, with a reduction at 2 weeks and a resolution at 6 months [82].

CMR is a fundamental tool for visualizing intracavitary thrombus formation in MI patients (Figure 1), which is a serious complication causing stroke or systemic thromboembolism, even when the acoustic window is limited and the echocardiography can give uncertain results. As a result, it has been demonstrated that an echocardiography has a sensitivity of 50% and a specificity of 100% if compared with a CMR in the detection of LV thrombi [11]. Another study reported that non-contrast echocardiography had a sensitivity of 33% and a specificity of 91% if compared to CMR in patients with LV dysfunction [83]. Meanwhile, contrast echocardiography had a higher sensibility and specificity if compared to non-contrast echocardiography (respectively, 61% vs. 33% and 92% vs. 82%, *p* < 0.05). A close agreement between contrast echo and cine CMR in the diagnosis of thrombus was found (κ = 0.79, *p* < 0.001), even if CMR was superior for thrombus prevalence, especially for those that were mural in shape or small in volume [84]. Serial CMR examinations should be recommended because ischemic patients with LV adverse remodeling have a higher incidence of developing late LV thrombosis (especially apical). Anterior wall involvement during MI, LV dysfunction, MVO, and adverse remodeling are more prevalent in patients with LV thrombi, which may represent a high-risk cohort to follow up for targeted screening with CMR [11].

### 4.2. T1/T2/ECV Mapping

Furthermore, since their introduction, myocardial mapping techniques have shown to identify the infarcted area during the subacute phase as areas of increase in T1 and T2 values [85], without the need of contrast agents (Figure 2). T1 mapping imaging has proven to be a valuable tool to delineate the infarcted area, the area at risk (AAR), and the non-infarcted myocardium [86]. In particular, native T1-mapping (nT1) performed as well as T2-mapping in delineating the edema-based AAR, whereas post-contrast T1-mapping can quantify IS. In parallel, the role of myocardial relaxometric values as potential predictive biomarkers of recovery or remodeling was also investigated. Dall’Armellina et al. [87] demonstrated the ability of T1 mapping to differentiate between reversible and irreversible myocardial injuries and their value as determinants of long-term LV recovery. Indeed, it has been demonstrated that acute T1 values in the scar area were higher if compared with remote myocardium and that higher acute nT1 values were predictive of a lower segmental function improvement at 6-month FU [87]. Other authors used different T1 value thresholds on the nT1 maps to detect irreversible damaged tissue and to predict LV remodeling at 6-months FU [88]. In acute MI patients, other evidence suggests that the zone with a reduced nT1 value within the infarcted area on T1 maps may depict the MVO, therefore correlating with adverse LV remodeling [7]. Although previous studies have focused mainly on the T1 mapping technique’s accuracy in identifying territories of myocardial edema and irreversible damage without contrast agent administration, T2 mapping has emerged as a robust and valuable alternative. The persistent T2 hyperintensity, defined as 2 SD above the signal in the remote zone at 6 months FU, was significantly associated with the initial STEMI severity, LV adverse remodeling, and poor long-term outcomes (*p* = 0.004) [89].

The 68Ga-pentixafor and 68Ga-DOTA-ECL1i PET imaging studies, respectively, showed that upregulation of CXCR4 and CCR2 chemokine receptors plays a role in adverse remodeling due to stimulation of myocardial inflammation and fibrosis pathways, offering new opportunities for personalized target therapies [18].

The extracellular volume fraction (ECV), calculated by combining native and post-contrast T1 values, is a surrogate measure of microscopic myocardial fibrosis and is an independent marker of prognosis in different cardiovascular diseases. It may play a role in the characterization of the post-reperfusion myocardial injury or in the detection of myocardial fibrosis in remote/non-infarcted myocardial regions [90]. The higher ECV values of remote myocardium (defined as the AHA segment 180 degrees from the infarct territory with normal wall motion and no LGE), acutely measured in reperfused STEMI patients, have been proved to be significantly associated with adverse LV remodeling, independently of remote-T1 values [90]. Similarly, an increased ECV of the infarcted myocardium area has been associated with a lower recovery of contractile function in affected segments [90]. The changes in segmental ECV values between the acute phase and 3-month FU (Δ-ECV) showed a significant difference among normal, edema, and infarcted segments (+0.8 vs. −1.8 vs. −2.9, respectively), reflecting the different tissue repair mechanisms. Further, normal segments demonstrating increased Δ-ECV showed deterioration in wall thickening and contractile function at FU [91]. It was even proved that ECV in acute reperfused MI could have higher accuracy than LGE extent to predict improvement of wall motion (AUC 0.77 vs. 0.66; *p* = 0,02); moreover, acute ECV was a better predictor of convalescent EF and attenuated strain in the infarct zone if compared to LGE [92].

## 5. 4D Flow Imaging: Intracavitary Blood Flow and Hemodynamic Forces

4D flow imaging, obtained by a three-directional velocity encoding phase contrast sequence, is an emerging technique that allows to assess LV intracavitary blood flow in three dimensions, thus representing a reproducible tool to study LV hemodynamics. 4D Flow Imaging enables the in vivo quantification of spatiotemporal 3D blood flow velocity (Figure 3) [93]. The velocity data can be used for non-invasive investigation of LV hemodynamics, which have already been characterized in terms of flow component subdivision [94], blood energetics [95], pressure gradients [96], and hemodynamic forces (HDF) [97].

LV blood flow kinetic energy (KE) assessment by 4D flow CMR has demonstrated clinical utility in LV diastolic assessment, prediction of thrombus formation, and adverse LV remodeling after MI [98]. The KE analysis demonstrated excellent reproducibility, which is important in the clinical translation of this novel imaging biomarker for hemodynamic assessment [99]. In ischemic heart disease (IHD) patients, it has been reported that intracavity blood flow energy is altered [99]. In a prospective study of 108 IHD patients, LV flow KE mapping obtained by 4D flow imaging was able to characterize flow changes, distinguishing those with LV thrombus from those without, by highlighting a delayed wash-in due to the higher intraventricular pressure gradients [95].

Due to the mutual interaction between myocardial wall mechanics and intracardiac fluid dynamics, 4D flow-derived biomarkers could deepen knowledge of LV physiology and elucidate the mechanisms of cardiac remodeling associated with specific patterns of LV dysfunction. Specifically, hemodynamic forces (HDFs) are exploited to quantify the load exerted by the intracavitary blood on the myocardial wall due to blood pressure gradients generated by the cyclical contraction and relaxation of the LV myocardium [100]. As a time-dependent force, whose magnitude is physically expressed in Newton, the single HDF vector is generally decomposed along three mutually orthogonal and anatomically relevant directions: (i) basal-apical (B-A), perpendicular to the atrioventricular plane and aligned with the LV long-axis; (ii) septal-lateral (S-L), parallel to the LV outflow tract view; and (iii) inferior-anterior perpendicular to the previous directions. Under physiologic conditions, the HDF vector is predominantly aligned with the basal-apical direction to optimize both blood flow ejection and diastolic LV chamber filling [97].

In previous studies, heterogeneous cohorts of patients with HF and LV mechanical dyssynchrony exhibited significantly altered HDFs compared to normal subjects [101]. In a group of patients with idiopathic dilated cardiomyopathy, HDFs were more heterogeneous compared to healthy volunteers in both direction and magnitude and acted to a greater extent orthogonally to the main flow directions [102]. More recently, HDFs have been quantified in ischemic heart disease (IHD) patients with adverse LV dilation and remodeling due to a previous anterior MI [100]. These patients revealed an almost absent basal-apical HDF component, which reduced on average by approximately one-half vs. healthy volunteers. Furthermore, a reduction in the septal-lateral HDF component was noticed, and the inferior-anterior HDF component was barely present. Consequently, these alterations enhance the relative magnitude of transverse (i.e., inferior-anterior and septal-lateral) forces with respect to the longitudinal (i.e., basal-apical) force, with the resultant HDF being more orthogonal to the main flow direction and exerting a larger force on the LV myocardial wall. Hence, HDF quantification confirmed the link between derangements in intracavitary hemodynamics and depressed LV function. Further, though it has been tested on a cohort of IHD patients with large, scarred regions and severe LV remodeling, HDF assessment could play a role in predicting the progression of adverse LV remodeling. Indeed, physiological intracavitary LV pumping is associated with a consistent HDF pattern; its derangements in both timing and magnitude may be promising and sensitive markers of ventricular dysfunction [97]. In this perspective, a recent study demonstrated that lower baseline systolic S-L HDFs and a higher diastolic S-L/B-A HDFs ratio during the subacute phase were associated with adverse remodeling at 4 months’ FU in a cohort of 49 STEMI patients. In addition, while systolic HDFs are altered by the direct consequence of myocardial damage on contractile impairment, diastolic HDFs reflect asynergy and asynchrony in LV wall motion. An increased diastolic S-L/B-A HDFs ratio may be the consequence of disproportionately high diastolic S-L HDFs, caused by stiffening of the infarcted wall, and low B-A HDFs, which reflect the reduction of the global elastic recoil effect, causing active suction of blood from the base to the apex [103].

The HDFs, as well as other 4D flow-based markers, may complement CMR-based assessment of LV global function, such as LV volumes, ejection fraction, and regional myocardial strain [104]. However, the potential of 4D flow-derived HDFs as early predictors of adverse LV remodeling and their advantage over myocardial CMR-based strain analysis still require further investigation through prospective longitudinal studies.

## 6. CMR in Candidates for Surgical Ventricular Restoration

LV remodeling in chronic MI is related to an increase in the EDV due to the presence of scar tissue, which progressively increases the risk of cardiac mortality, leading to cardiac transplantation in some cases [105]. A surgical technique to slow down this remodeling process was introduced by Vincent Dor in 1989 and consists of surgical ventricle restoration (SVR), a treatment option for patients with post-ischemic akinetic LV dilation, aiming at reducing LV volume and excluding the scar from the cavity [106]. The effect of this procedure is an improvement in LV function and clinical status [107].

CMR imaging has already proven to be a highly reliable, non-invasive imaging technique for the evaluation of SVR candidates [108]. The evaluation of LV volumes and extension of akinetic myocardium is crucial to planning the appropriate surgical procedure and correctly normalizing the shape of the LV. In CMR, accurate quantification of both LV EDV and ESV is mandatory before surgery. A long-term prognosis in these patients is determined by the relationship between accurate methods for measuring ventricular volumes and the extent of SVR volume reduction. SVR reported a >33% reduction in ESVi followed by an 8-year survival > 80% when ESVi < 90 mL/m^2^. Conversely, an inadequate volume reduction of 15% results in 100% mortality at 8 years if the ESVi is >90 mL/m^2^ [109].

The scar extension estimation by the LGE technique allows discrimination of healthy tissue from infarcted or nonviable myocardial tissue. This is important in SVR planning because when the viable myocardium cannot be directly sutured, a patch is positioned and the scar is sutured over the patch. The excluded wall is closed over viable myocardium or the patch to reinforce the suture [110]. As a result, an LGE atypical distribution is observed after the procedure [111]. The characterization of scar tissue interferes with the post-surgical outcome. In particular, the presence of LGE in the antero-basal LV segments was the only independent negative predictor of outcome [112]. A T1 mapping analysis with ECV estimation could be useful to determine the status of the myocardium without LGE. In patients with non-ischemic dilated cardiomyopathy, an anteroseptal abnormal ECV in the presence of LGE seems to increase the patient’s risk [113].

A recent study by Solowjowa et al. [114] on 205 patients undergoing SVR demonstrated the viability of CCT for surgical planning and FU as an alternative to CMR due to its capability of exact true dilation/aneurysm volume detection combined with short examination times and a lack of technical restrictions in critical patients (e.g., patients with CIEDs). The limitations of the use of CCT in this context, compared to CMR, are represented by an inferior accuracy in scar transmurality detection, viability assessment of the remaining myocardium, and strain analysis due to the low temporal resolution, while exposing the patient to a high radiation burden.

CMR imaging, then, is an adequate technique to evaluate patients before and after surgical ventricle restoration, providing a comprehensive assessment of anatomy and giving anatomical, functional, and tissue viability information (Figure 4).

## 7. Conclusions

LV adverse remodeling is a complex biological process beginning early after MI and increasing the risk of long-term poor cardiovascular outcomes. Additionally, CMR imaging is a valuable tool in estimating adverse remodeling by accurately assessing LV volumes and myocardial deformation and by exploring myocardial tissue features (Figure 5). Further, STEMI patients presenting an anterior location, LV dysfunction, or LV aneurisms should undergo both a baseline and follow-up CMR study. A pre-discharge CMR should be indicated in those with an echocardiography LVEF < 50%. On the other side, a period ≥3 months after STEMI should be sufficient to obtain a precise quantification of scar burden and LVEF recovery. Thus, CMR has been proven to be a very valuable tool to study myocardial viability and to select ischemic patients eligible for coronary percutaneous revascularization and surgical ventricular restoration.

## Figures and Tables

**Figure 1 jcm-12-00334-f001:**
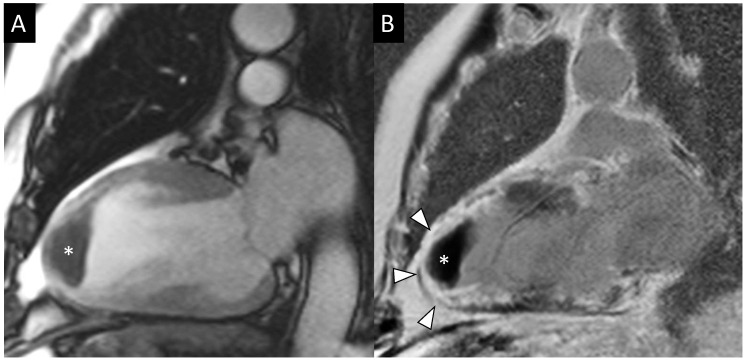
Apical post-infarction aneurysm and intracavitary thrombus. The cine-MR image acquired on the vertical long-axis view (**A**) showed aneurysmal remodeling of the apex with a homogenous hypointense thrombus adhered to the wall (*, size of 4 × 2.5 cm). On the LGE image (**B**), the mass appeared markedly hypointense compared to the ventricular blood pool and was surrounded by an enhanced infarcted LV wall (transmural LGE pattern, arrowheads).

**Figure 2 jcm-12-00334-f002:**
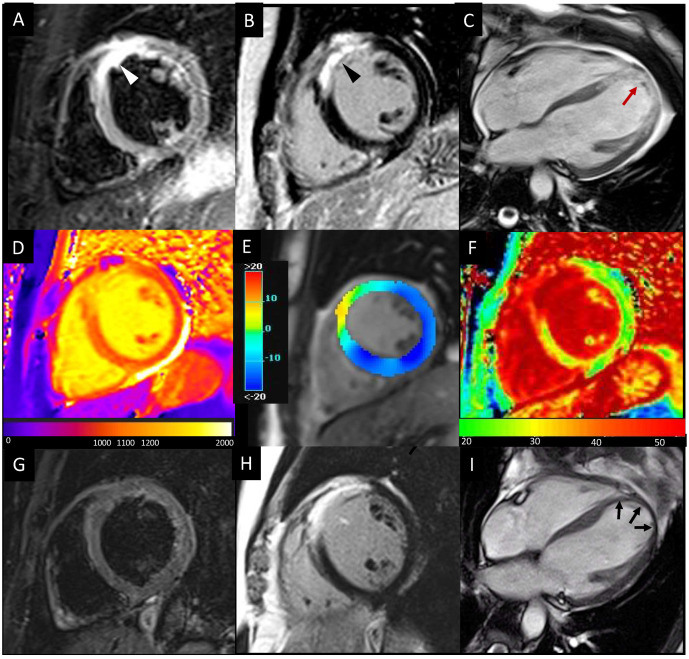
Cardiac magnetic resonance images of LV anterior myocardial infarction due to left anterior descending artery occlusion, acquired during the subacute phase (**A**–**F**) and six-month FU (**G**–**I**). T2-w STIR (**A**) and LGE (**B**) images acquired on the mid-ventricular short axis view show myocardial edema (white arrowhead) and necrosis (black arrowhead), respectively, of the antero-septal wall. Cine MR images acquired in four-chamber view (**C**) show thickening of the damaged LV apical wall due to post-ischemic edema and the presence of a tiny apical thrombus (red arrow). Areas of increased myocardial T1, reduced circumferential strain (CS), and expansion of extracellular volume fraction (ECV) well match the infarcted region, respectively, on the native nT1 (**D**), CS (**E**), and ECV (**F**) maps. FU images demonstrate disappearance of myocardial edema on STIR images (**G**), persistence and shrinkage of the scar on LGE images (**H**), and adverse LV remodeling with thinning of the infarcted walls and rounding of the apex as for aneurysmal evolution (black arrows, **I**). LV: left ventricle; STIR: short tau inversion recovery; LGE: late gadolinium enhancement.

**Figure 3 jcm-12-00334-f003:**
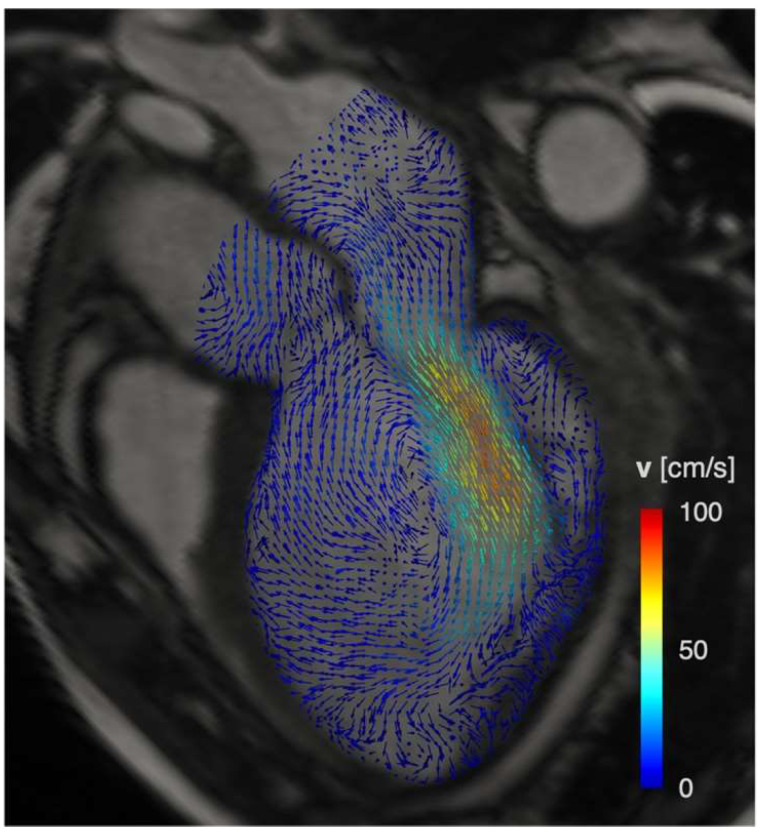
A color-coded 4D flow vector map at mid-diastole demonstrates the intracavitary flow with the generation of a large mid-ventricular vortex in patients with post-ischemic LV dilation.

**Figure 4 jcm-12-00334-f004:**
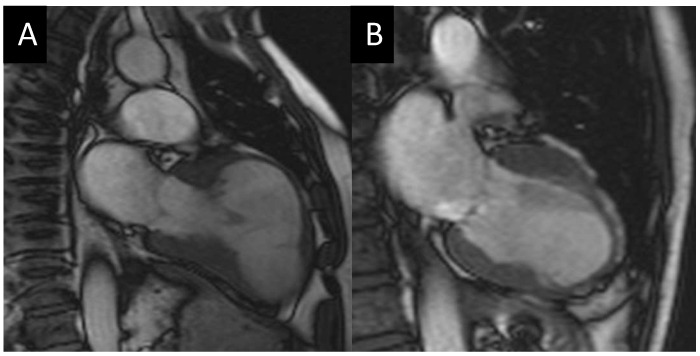
Cardiac MR imaging of a 72-year-old man with an occluded left anterior descendent coronary artery treated with a by-pass and surgical ventricle restoration. Two-chamber cine true fast imaging with steady-state free precession sequence in the end-systolic phase before (**A**) and six months after surgery (**B**).

**Figure 5 jcm-12-00334-f005:**
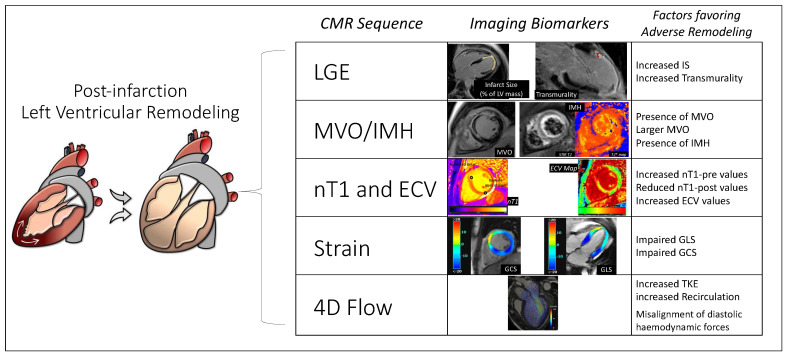
CMR imaging biomarkers in the risk prediction of adverse remodeling.

## Data Availability

Not applicable.

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
