# Peer review of "Left Ventricular Adverse Remodeling in Ischemic Heart Disease: Emerging Cardiac Magnetic Resonance Imaging Biomarkers"

_jcm, 2023, doi:10.3390/jcm12010334_

Round 1

Reviewer 1 Report

1.       Besides cardiac magnetic resonance (CMR) imaging to define left ventricular (LV) remodeling, please add the introduction of other technologies for defining LV remodeling in the first section of background introduction, and compare their advantages and disadvantages with CMR in the main text.

2.       Line 275, please add one space between “CMR” and “[11]”. The same in line 417, line 462.

3.       Line 335, please double check the figure number in the bracket “(G-J)”, is it “(G-I)”?

Author Response

Thanks to the reviewer for his suggestions.

Point 1 We added a paragraph about other technologies for defining LV remodeling such as 2D and 3D echocardiography, 2D speckle tracking echo, cardiac CT , SPECT and nuclear imaging in the Introduction section: page 2 lines 83-100 and page 3 lines 114-121; and then, we discussed the above mentioned technologies comparing their advantages and disavantages with respect to CMR in different sections in the main text: in section "2. LV remodeling volume-based definition" page 3 lines 129-145 , in the section "3. Myocardial strain analysis" page 4 lines 205-209 and 212-217, page 5 lines 229-234 and lines 252-256 ,in the section "4. Tissue characterization, subsection 4.1 LGE, MVO and IMH"  page 6 lines 293-297, page 7 lines 378-381, page 8 lines 410-420, in the section "4.2. T1/T2/ECV mapping" page 9 lines 466-470, and in the section "6. CMR in candidates to surgical ventricular restoration" page 13 lines 627-636

2 and 3 points amended as requested

Reviewer 2 Report

Dear Authors,

I read the paper entitled 'Left ventricular adverse remodeling in ischemic heart disease: emerging cardiac magnetic resonance imaging biomarkers’ with great interest.

This paper summarizes important topic regarding post-ischemic left ventricular (LV) remodeling and usefulness of cardiac magnetic resonance (CMR) imaging in its assessment. The title describes the core message of the paper. The abstract incorporates key messages, in a concise manner. The structure of the paper is accurate. Importantly, paper draws attention that new CMR quantitative biomarkers emerged as predictive of post-ischemic adverse remodeling, such as T1 mapping, myocardial strain and 4D flow.

However, I have some suggestion regarding this paper. It will be valuable to add some information about usefulness of CMR in the assessment of LGE on the posterolateral wall of the LV and thus in the qualification of patients for implantation of cardiac resynchronization therapy (CRT). 

Author Response

Thanks to the reviewer for this suggestion.

We added a paragraph in the section “4. Tissue characterization, subsection 4.1 LGE, MVO and IMH”  Page 6 lines 293-297 regarding LGE in the postero-lateral wall and CRT as follow: “An important role of LGE in patients with post ischemic heart failure is also related to the stratification of patients with an indication to conventional cardiac resynchronization therapy (CRT). In literature it is reported that a transmural LGE in the posterolateral wall is related to nonresponse to CRT. Then this evidence shows that CRT does not reduce LV dyssynchrony in patients with transmural scar tissue in the posterolateral LV segments, resulting in clinical and echocardiographic nonresponse to CRT even if extensive LV dyssynchrony exists. A transmural scar tissue in the target region (for LV pacing) prohibits response to CRT.”

Reviewer 3 Report

This manuscript reviews the left ventricular adverse remodeling in ischemic heart disease and the emerging cardiac magnetic resonance imaging biomarkers, which has important clinical significance for the treatment of ischemic heart disease. 

Author Response

We thank the reviewer for his valuable comment